# Design, Implementation, and Empirical Validation of an IoT Smart Irrigation System for Fog Computing Applications Based on LoRa and LoRaWAN Sensor Nodes [note 1]

**DOI:** 10.3390/s20236865

**Published:** 2020-11-30

**Authors:** Iván Froiz-Míguez, Peio Lopez-Iturri, Paula Fraga-Lamas, Mikel Celaya-Echarri, Óscar Blanco-Novoa, Leyre Azpilicueta, Francisco Falcone, Tiago M. Fernández-Caramés

**Affiliations:** 1Department of Computer Engineering, Faculty of Computer Science, Universidade da Coruña, 15071 A Coruña, Spain; ivan.froiz@udc.es (I.F.-M.); o.blanco@udc.es (Ó.B.-N.); 2Centro de investigación CITIC, Universidade da Coruña, 15071 A Coruña, Spain; 3Department of Electric, Electronic and Communication Engineering, Public University of Navarre, 31006 Pamplona, Spain; peio.lopez@unavarra.es (P.L.-I.); francisco.falcone@unavarra.es (F.F.); 4 Institute of Smart Cities, Public University of Navarre, 31006 Pamplona, Spain; 5 School of Engineering and Sciences, Tecnologico de Monterrey, 64849 Monterrey, NL, Mexico; mikelcelaya@tec.mx (M.C.-E.); leyre.azpilicueta@tec.mx (L.A.)

**Keywords:** IoT, LP-WAN, LoRaWAN, LoRa, 3D-ray launching, fog computing, smart cities, wireless sensor networks (WSN), smart irrigation, sustainability

## Abstract

Climate change is driving new solutions to manage water more efficiently. Such solutions involve the development of smart irrigation systems where Internet of Things (IoT) nodes are deployed throughout large areas. In addition, in the mentioned areas, wireless communications can be difficult due to the presence of obstacles and metallic objects that block electromagnetic wave propagation totally or partially. This article details the development of a smart irrigation system able to cover large urban areas thanks to the use of Low-Power Wide-Area Network (LPWAN) sensor nodes based on LoRa and LoRaWAN. IoT nodes collect soil temperature/moisture and air temperature data, and control water supply autonomously, either by making use of fog computing gateways or by relying on remote commands sent from a cloud. Since the selection of IoT node and gateway locations is essential to have good connectivity and to reduce energy consumption, this article uses an in-house 3D-ray launching radio-planning tool to determine the best locations in real scenarios. Specifically, this paper provides details on the modeling of a university campus, which includes elements like buildings, roads, green areas, or vehicles. In such a scenario, simulations and empirical measurements were performed for two different testbeds: a LoRaWAN testbed that operates at 868 MHz and a testbed based on LoRa with 433 MHz transceivers. All the measurements agree with the simulation results, showing the impact of shadowing effects and material features (e.g., permittivity, conductivity) in the electromagnetic propagation of near-ground and underground LoRaWAN communications. Higher RF power levels are observed for 433 MHz due to the higher transmitted power level and the lower radio propagation losses, and even in the worst gateway location, the received power level is higher than the sensitivity threshold (−148 dBm). Regarding water consumption, the provided estimations indicate that the proposed smart irrigation system is able to reduce roughly 23% of the amount of used water just by considering weather forecasts. The obtained results provide useful guidelines for future smart irrigation developers and show the radio planning tool accuracy, which allows for optimizing the sensor network topology and the overall performance of the network in terms of coverage, cost, and energy consumption.

## 1. Introduction

The lack of water and water-stressed areas are becoming serious problems to many places around the world. According to the United Nations (UN), by 2025, water scarcity will affect 1.8 billion people and two thirds of the population will live under water-stressed conditions [1]. In addition, water demand is expected to grow 50% by 2050 due to the increase in the number of people living in urban areas. As a consequence, drought is expected to have a widespread short and long-term impact on the everyday life of many people. Together with that, climate change is increasing flood risks and extreme precipitation events [2].

To tackle the previous issues, the UN recommends taking actions on three aspects [1]: the development of drought monitoring systems, the assessment of the potential risks and vulnerabilities, and the design of drought risk mitigation strategies. The present article deals with the latter by designing, implementing, and validating empirically a smart irrigation system that can be deployed in urban and suburban scenarios.

Specifically, the irrigation system proposed in this article has been deployed on a university campus and makes use of Internet of Things (IoT) nodes able to collect environmental data (e.g., soil temperature/moisture and air temperature) and that can be activated remotely, either through manual commands or through intelligent automated actions. In addition, the proposed system overcomes the main communications challenges that arise in urban scenarios (e.g., the need for long-range communications, long battery life, or high network node capacity) by using Low-Power Wide-Area Network (LPWAN) transceivers based on LoRa and LoRaWAN, a standard that has recently attracted a lot of attention from industry and academia [3,4,5].

This article also deals with another emerging technology that has gained momentum: fog computing [6]. Fog computing is a type of edge computing [7,8,9] that allows for extending certain cloud computing features to the edge of the network through the use of low-power computational devices like Single-Board Computers (SBCs). Thus, low-cost fog computing devices have proven to be able to provide responses to IoT nodes significantly faster than a cloud [10], besides bringing location awareness to such responses (i.e., the responses provided by a fog device to the monitored IoT nodes can be adapted and refined depending on their physical location).

The optimization of the deployment of fog computing gateways and IoT nodes requires carrying out radio planning tasks. Such tasks are essential for scenarios like the one modeled for this paper (i.e., an university campus), which are specially challenging in terms of radio propagation due to their large dimensions and the numerous obstacles that impact electromagnetic waves. For that purpose, empirical models are usually employed, which give rapid results but, with high errors, since they are very site-specific. In order to obtain accurate estimations, in this work, an in-house developed 3D-Ray Launching (3D-RL) deterministic algorithm has been used to characterize the LoRa/LoRaWAN radio channel at 868 MHz and 433 MHz. As it will be later described in Section 4, the algorithm provides a good trade-off between accuracy and computational time. In addition, in order to complete the radio planning study of the scenario under analysis, and validate the obtained simulation results, empirical RF measurements were obtained for both 868 MHz and 433 MHz frequency bands. Specifically, the following are the main contributions of this article:

A comprehensive state-of-the-art section is provided on the most relevant academic smart irrigation systems and on the use of LoRa/LoRaWAN for underground scenarios.An architecture is proposed for building novel smart irrigation systems based on the deployment of LoRa/LoRaWAN transceivers and fog computing nodes.The implementation of the proposed system and its hardware is described in detail so as to allow future developers to replicate it easily.The radio planning analysis of the scenario (i.e., a university campus) is presented together with an empirical measurement campaign to corroborate the analytical results.

The rest of this article is structured as follows. First, Section 2 analyzes the most relevant previous works on smart irrigation systems and on the use of LoRa and LoRaWAN in underground scenarios. Section 3 describes thoroughly the proposed communications architecture, the deployment scenario, and the implementation of the different components of the smart irrigation system. Section 4 provides details on the performed radio analysis: its characteristics, the configuration of the used 3D-RL tool, and the obtained simulation results. Finally, Section 5 is dedicated to the description and analysis of the results obtained during an empirical measurement campaign, while Section 6 is devoted to conclusions.

## 2. State of the Art

### 2.1. IoT Smart Irrigation Systems

IoT has gained in the last few years a lot of interest as a key enabler technology in smart agriculture [11]. With respect to smart irrigation systems, there are a number of recent academic works focused on decision planning and operation support. For instance, Khan et al. [12] describe a Decision Support System (DSS) that uses Xbee devices to cover a relatively small area. Another interesting work was presented by Togneri et al. [13], who designed a flexible IoT platform that enables developing Machine Learning (ML)-based solutions for smart irrigation. Its core components are a set of software, hardware, and communication technologies (e.g., LoRaWAN) that include soil moisture sensor probes, the use of an open-source platform (FIWARE), a SPARQL event processing architecture, as well as specific services for irrigation planning and operation. Unfortunately, the mentioned article does not provide further details on the communications architecture. Recently, Boursianis et al. [14] proposed a sophisticated smart irrigation system that exploits the capabilities of 5G networks and energy harvesting. A different approach is proposed by Munir et al. [15], which uses an intelligent method based on a fuzzy logic to schedule irrigation based on different parameters. Since the design of smart irrigation systems involves several considerations, Table 1 shows an overview of the main aspects to be taken into account. A complete and detailed overview of the current state-of-the-art regarding the IoT irrigation system for precision agriculture is presented in [16]. The authors review in detail common sensors and actuators to develop the IoT irrigation systems, most common parameters to be monitored, the type of irrigation, the type of node, the type of wireless communication technology, among others.

### 2.2. Communication Technologies for Smart Irrigation Systems

Among the most popular LPWAN communications technologies (e.g., Sigfox, LoRa/LoRaWAN, WavIoT, Random Phase Multiple Access (RPMA), Narrowband IoT (NB-IoT) or LTE-M), LoRa and LoRaWAN were selected due to their expected communications range and performance in scenarios where significant signal attenuation occurs, like is the case for underground communications. In fact, some authors already performed thorough comparisons among LPWAN technologies and showed that RPMA and LTE-M incur additional path losses in comparison to Sigfox or LoRAWAN technologies [17]. In the case of LoRa or LoRaWAN, they use a proprietary Chirp Spread Spectrum (CSS) modulation in the PHY layer, but their MAC layer is an open specification. On the contrary, Sigfox makes use of an open chipset, but its network is closed (i.e., it is necessary to pay a periodic fee). Nonetheless, it must be noted that there is no fully open LPWAN protocol stack (for PHY and MAC layers).

To provide coverage for large areas, LoRa/LoRaWAN solutions have been suggested for developing smart irrigation systems. For example, Gloria et al. [18] describe a Wireless Sensor Network (WSN) controlled by a single gateway (a broker) that is in constant communication with an online server that uses LoRa peer-to-peer connections. Message Queuing Telemetry Transport (MQTT) via a WiFi connection is used for exchanging messages between the server and the nodes. Another solution was presented by Usmonov et al. [19], who detailed a LoRaWAN-based cost-effective wireless control system for drip irrigation. Such a solution uses a master station to relay packets between the control application and the deployed end-device nodes. It is also worth mentioning the work described in [20], which details the design and implementation of a LoRaWAN smart irrigation system in a typical urban environment. In such a work, the authors indicate that the communications distance between the irrigation node and the gateway can reach up to 8 km (covering an area of up to 2 km^2^) and provide energy-consumption results that consider different operating modes of the end nodes. Note that larger areas could be easily covered with LoRa. Finally, it is relevant to mention the work of other authors like Citoni et al. [21], who studied the deployment limitations of large-scale LoRaWAN IoT applications. Specifically, such a work analyzed the impact of packet collision on the scalability of a LoRaWAN deployment.

### 2.3. LoRa and LoRaWAN in Underground Scenarios

In this article, underground scenarios are defined as the ones that make use of wireless communications transceivers to send and to receive data from under the soil. Such transceivers have to be able to avoid the communications issues that arise when operating underground in a number of industrial applications, like maintenance [22], mining [23,24], agriculture [25,26], or structural engineering [27].

In underground scenarios, the radio signal absorption loss caused by the soil severely affects the performance of traditional wireless communications technologies. In particular, factors that affect signal propagation in underground scenarios are the soil type and density, the soil moisture, the existence of certain elements (e.g., plant roots, rocks or underground pipelines), and the burial depth [28].

Although authors like Akyildiz et al. [29] suggest some interesting preliminary results, the existing research on in-soil underground communications is still not mature. For instance, one of the first empirical studies on the topic was presented in 2010 by Silva et al. [30]. In such a work, the authors tested ZigBee MICA2 and MICAz motes in an agriculture field. Their results showed a high asymmetry between underground and above ground communications, being highly dependent on burial depth and soil moisture (e.g., a 21% increase in water content decreased the signal coverage by more than 70%). Besides ZigBee, other authors have tested in similar scenarios some of the most popular traceability technologies like Radio Frequency Identification (RFID) [31,32] for developing underground applications [33,34].

LoRa and LoRaWAN make use of a chirp spread modulation that offers several advantages for underground communications [35]: better coverage than technologies like WiFi or ZigBee, a sensitivity on the order of –130 dBm, low energy consumption, and low cost. There are a number of recent papers that study the performance of LoRa/LoRaWAN in underground scenarios. For instance, Wan et al. [36] designed and evaluated a LoRa propagation test node. They concluded that, if rainfall and irrigation happen frequently, their associated channel losses need to be compensated with increased transmission power or with the use of high-gain antennas, the latter being the recommended approach. Moreover, the authors provide additional recommendations: with respect to the burial depth of the node, it should be as close as possible to the surface and, regarding the Packet Success Rate (PSR) of underground communications, a shorter LoRa payload is required for a higher PSR.

LoRa in-soil propagation was also studied by Xue-fen et al. [37], who proposed a smartphone-based LoRa underground measurement system. Thus, the authors described thoroughly two scenarios (a farmland and a garden) that considered the type of soil (e.g., homogeneous loam) and plants (e.g., mainly rice), the root depth of grass and camphor trees, and the characteristics of underground iron pipelines that are used for municipal drainage and communications. They also considered two different paths: underground-underground (UU) and underground-above ground-underground (UAU). In their experiments, the attenuation of the radio signals in the UU path, which is related to the soil characteristics and to other underground obstructions, was higher than for the UAU path. Thus, the actual attenuation in the analyzed environment needed to be obtained through measurements, due to the randomness of the soil and the obstructions. Xue-fen et al. also note the negative impact of soil moisture on LoRa in-soil propagation considering the precipitation rate, the randomness of rainfalls, and the amount of irrigation water in the area for the last 12 h. Since the water content in deep soil is usually higher than the one of the surface, the packet success rate decreases with deeper burial depth. A similar approach was taken in [38], where, after developing an in-soil propagation test node, the authors performed measurements and analyses of the results in a riverside park. In the case of Lin et al. [39], they focused their research on an experimental analysis on the influence of various physical layer parameters (e.g., spreading factor, coding rate, bandwidth) on LoRa’s propagation performance.

### 2.4. Key Findings

After reviewing the state-of-the art, it can be concluded that there are a number of recent articles that make use of LoRa/LoRAWAN as preferred wireless technology for IoT smart irrigation systems. In contrast to the proposed system, previous research is mainly focused on DSS capabilities introducing statistical or ML/AI (Artificial Intelligence) techniques for decision-making, or on the proper irrigation system design, focused on optimizing hardware capabilities. Thus, there is barely any system that proposes the use of fog or edge computing. In addition, in relation to radio signal absorption loss in underground scenarios, most authors recommend performing in-soil wireless propagation measurements before the design, installation, and maintenance of any smart irrigation system, but the use of scenario simulators together with practical deployments has been barely documented.

Table 2 compares the main features of the most relevant smart irrigation systems. As it can be observed, in contrast to the compared systems, this article presents an IoT LPWAN smart irrigation system with a fog computing-based architecture whose node and gateway locations can be optimized for urban areas thanks to the help of an in-house developed 3D-ray launching radio planning simulator.

The system proposed in this article is based on the use of LoRa or LoRaWAN due to their ability to provide medium and long-range coverage in both indoor and outdoor scenarios, thus fitting into the traditional areas where smart irrigation is required. Specifically, previous literature evaluated LoRa range, obtaining up to 30 km for boat communications and 15 km for car communications [40], up to 8 km (but losing many packets) and up to 4 km (losing a few packets) within urban environments [41], up to 500 m for very dense-vegetation environments [42] and a radius coverage of 1.8 km for a soil monitoring system in agriculture by means of LoRa devices with worse performance in terms of sensitivity than the ones used in this article [43].

Besides range, LoRa and LoRaWAN have other features that are attractive for smart irrigation deployments: their energy consumption is relative low (RX current is usually around 10 mA, while deep-sleep current is under 200 nA), its robustness against interference has been proven [44,45], transceiver cost is also low (less than US$ 8, as of writing) and there is no need to pay monthly fees to a mobile carrier. Nonetheless, developers must note that LoRa and LoRaWAN have not been devised to transfer large data payloads fast (the theoretical maximum data rate is usually 50 Kbps for LoRa and 27 Kbps for LoRaWAN) and they make use of Industrial-Scientific-Medical (ISM) bands, so interference may arise from other devices that operate in the same radio frequency.

## 3. Design and Implementation of the System

### 3.1. Smart Irrigation Scenario

Before describing the design and implementation of the proposed smart irrigation system, it is first necessary to detail the characteristics of the scenario where it is deployed in order to understand the choices made. Such a scenario is located in the Campus of Elviña, at the University of A Coruña (Spain), in an area that covers 7500 m^2^ (with 50 m of width and 150 m of length). The specific area is shown in Figure 1 together with the locations of the deployed smart irrigation nodes (red circles, numbered 1 through 8) and communication gateways (purple circles, numbered 9 to 11).

The area where the smart irrigation system was deployed includes elements typically found in urban and suburban environments: besides the gardens to be irrigated (composed by lawn and diverse types of trees), there are several buildings, roads, and cars. As an example, the pictures in Figure 2 show the areas where smart irrigation nodes 1, 3, and 4 were located, and where the mentioned elements of the scenario can be observed. It is important to note that the size of the scenario and the specific area were selected for convenience reasons, as they are close to the lab of the researchers, and they are representative of the entire campus. In such a scenario, gateways and nodes were deployed in an area that is large enough to validate the proposed communications architecture and the smart irrigation system, but larger areas could be easily covered by adding more LoRaWAN nodes and, if needed, additional gateways. Moreover, the scalability provided by LoRa for the communication between nodes and the gateway can be easily replicated by the used 3D-RL simulator, which, once the scenario to be simulated is created, only requires specifying the desired number of nodes and their location in order to obtain simulation results. Nonetheless, it must be emphasized that the higher the number of nodes, the higher the computational cost/time that will be required for the simulations.

### 3.2. Designed Communications Architecture

Figure 3 shows the communications architecture of the proposed system. In such a figure, the different components are connected through arrows that indicate the way communications are performed (e.g., the two-way arrow that connects Remote Services with Third-Party Services indicate that the former make requests to the latter to ask for certain information and the latter respond with the requested data). Specifically, the proposed architecture is composed of three layers:

IoT Node Layer. This layer consists of smart irrigation IoT nodes that exchange information with local gateways. The double arrows that communicate the IoT Node Layer with the Fog Computing Layer in Figure 3 represent bidirectional communications, which indicate that remote commands can be sent to the IoT nodes, while such nodes can send information about the successful execution of the commands (e.g., in order to determine whether the system works properly) or on the state of their sensors.Fog Computing Layer. This layer is composed by local gateways distributed over different locations to extend network connectivity across large areas. Such gateways provide redundancy, low-latency responses and distributed processing, thus off-loading tasks from the remote cloud.Remote Service Layer. This layer is located in the cloud and collects data from the deployed components of the smart irrigation system. The collected data can be processed and stored on the cloud database in order to be later shown to remote users through a user-friendly interface. Moreover, the services on the cloud can exchange information with useful third-party services (e.g., an external weather forecast service).

### 3.3. LoRa and LoRaWAN

The LoRa physical layer operates in the ISM bands (e.g., 868 MHz, 433 MHz, and 915 MHz). In Europe, only the 868 MHz and 433 MHz bands can be used. Transmitted power is limited to 14 dBm Effective Isotropic Radiated Power (EIRP) with a 1% duty cycle limit of on-air time. LoRa integrates a Forward Error Correction (FEC) to increase robustness against noise and burst interference. Transmissions can be sent on different spreading factors, and, depending on the modulation and transmission power, the link budget can be as high as 155 dB.

LoRaWAN involves a protocol stack with LoRa as the physical layer. It uses Adaptive Data Rate (ADR) for managing individual data rates and maximizing the battery lifetime of each connected device. The LoRaWAN network architecture has a star topology in which end nodes communicate with gateways which in turn connect to network servers. A summary of LoRa/LoRaWAN main characteristics can be seen in Table 3. In addition, the interested reader can obtain further details in [46].

### 3.4. Implemented Communications Architecture

To deploy the proposed system making use of LoRa/LoRaWAN, the three different layers defined in Figure 3 were implemented as illustrated in Figure 4. Note that Figure 3 and Figure 4 show an identical Remote Service Layer, but they differ significantly on their fog computing layer and on the way they manage IoT nodes. Regarding Figure 4, it is worth pointing out that, for the sake of clarity, it only refers to LoRaWAN, since a LoRa-based architecture would be a subset. The main difference between a LoRa-based and a LoRaWAN-based architecture is that LoRa does not need a specific LoRa Server, while, in the case of LoRaWAN, such a server is necessary to manage different aspects of the LoRaWAN network, like its security or certain packet management tasks. For such a reason, the rest of this subsection and the next would only refer to LoRaWAN. Considering the previous clarification, the layers depicted in Figure 4 were implemented as follows:IoT Node Layer. This layer is composed of IoT nodes that embed LoRaWAN transceivers, soil sensors, and irrigation actuators. The LoRaWAN packets sent by the IoT nodes can be collected by one or more nearby LoRaWAN gateways.Fog Computing Layer. In this layer, LoRaWAN gateways collect packets from the deployed smart irrigation nodes and then send them to a central LoRaWAN server, where they are decoded and processed in order to provide fog computing services.Remote Service Layer. It works as it was previously described in Section 3.2.

### 3.5. Implemented IoT Node Layer

The implemented LoRaWAN IoT nodes send and receive information using the LoRaWAN protocol. Specifically, the deployed nodes are able to send periodically the data collected from their soil temperature/moisture and air temperature sensors. In addition, the nodes can receive remote commands to carry out different tasks (e.g., to activate the irrigation actuator or to establish the watering schedule).

It must be noted that LoRaWAN end devices can be divided into three classes (A, B, and C) depending on their needs regarding downlink communications (i.e., for transmitting data from a LoRaWAN gateway to the irrigation nodes) [47]. In order to reduce node stand-by power consumption, the smart irrigation nodes were chosen to be of class A. Such a class allows the nodes to remain asleep most of the time to save power and to receive commands only during a short period of time after sending data (this fact implies that remote commands will not be executed in real time).

Each IoT node was designed so that different sensors can be used. For illustration purposes, Figure 5 shows one of the smart irrigation IoT nodes disassembled. As it can be observed, each node includes interfaces for a DS18820 temperature sensor [48] and an SHT15 temperature and moisture sensor [49]. In addition, each node provides an interface to connect a 12 V-DC solenoid valve (Seafront G763hwmgsk) to control water irrigation.

The IoT node main board is based on a Heltec LoRa 32 V2 [50], which provides WiFi connectivity, an OLED Display, and an SX1276 LoRa module. Moreover, a Real-Time Clock (RTC) was embedded so that the microcontroller can take time-accurate actions when the network time is not available (i.e., based on the local time indicated by the RTC).

It is possible to make use of the characteristics of the previously described hardware to determine the performance of the system. Specifically, the characteristics of the selected valve allow for estimating that its water flow is about 8 L per minute (a more precise estimation could be calculated by embedding a water flow meter in every IoT node, but this will also derive into the development of more expensive and power hungry nodes). Assuming that the approximate amount of water required by the lawn to stay green is about 10 L per square meter per day, it is possible to calculate the savings yielded by the proposed smart irrigation system.

For calculating water savings, it is necessary to consider that the smart irrigation system can irrigate depending on the weather, since the system will not activate the node valves if the weather forecast indicates that it is going to rain. Thus, if the rainfall prediction is higher than 10 L per square meter, it is not necessary to irrigate throughout the whole day. On the contrary, if sunny weather conditions are expected, with high solar radiation and low humidity, water evaporation increases and, as a consequence, the amount of required irrigation will be higher.

The proposed system allows, by means of automatic planning systems or by manual programming, to reduce the amount of water to the minimum necessary when weather conditions are adequate and to increase it when soil is too dry, thus making a more efficient use of water and thus reducing the waste of water considerably.

Specifically, considering only weather forecasts and by using local forecasts data (provided by a regional government agency called Meteogalicia [51]) to obtain average rainfall over a year in the selected irrigation area, it has been estimated that 22.8% water savings would be achieved in comparison to a fixed-schedule irrigation system. Such an estimation comes from the results shown in Figure 6, which depicts the amount of water used throughout a whole year (from November 2019 to November 2020) by the smart irrigation system (red line) depending on the amount of rainfall (blue line). As an additional example, Figure 7 shows a more detailed graph for the month of September 2020 in order to illustrate the difference in water usage depending on weather conditions. As it can be clearly observed in Figure 7, the smart irrigation system makes use of roughly 10 L per square meter when there is no rain, but such an amount decreases in proportion to the rainfall, thus saving a significant amount of water.

### 3.6. Implemented Fog Computing and Remote Service Layers

The higher layers of the architecture depicted in Figure 4 were implemented as follows:Fog Computing Layer. It makes use of a central LoRaWAN server and LoRaWAN gateways that can be scattered throughout large areas. The LoRaWAN gateways are based on a regular microcontroller (STM32L1), while the LoRaWAN server can be executed on a regular PC. When IoT nodes need to send data, they send a LoRaWAN packet that will be received by the nearest gateway. Then, the gateway forwards the packet to the LoRaWAN server in the local network, which can send it later to the cloud, where all the information from the different LoRaWAN networks is aggregated. The LoRaWAN server is also able to perform local actions based on the received information from a specific area before interacting with the cloud.Remote Service Layer. The core of the Remote Service Layer manages data collection through Node-RED [52]. In addition, a MongoDB database [53] is used to store the collected data. Furthermore, this layer is able to make use of third-party services like weather forecasters (for deciding irrigation schedules), which can be easily integrated with the Remote Service Layer through Representational State Transfer (REST) Application Programming Interfaces (APIs).

### 3.7. Enabled Applications

The proposed smart irrigation system provides numerous advantages with respect to traditional systems thanks to the intelligent automation of the irrigation process. Specifically, the following are some of the most relevant features of the proposed system that enable the development of advanced irrigation applications:Irrigation scheduling can be adjusted dynamically and in a smart way by considering multiple information sources like weather forecasts.The system can adjust dynamically to changing environmental conditions, like soil moisture, ground temperature, or real-time weather conditions.The system is able to adjust the irrigation schedule dynamically so as to adapt it to the species that are grown on each individual green area.In case of having directional irrigators on the IoT nodes, it is possible to establish specific dynamic irrigation patterns with the objective of watering very specific areas.The system can be easily scaled (it autonomously connects to the nearest gateway), thus being able to cover large areas thanks to the use of LPWAN technologies.It is straightforward to add additional IoT sensor nodes (which do not need to embed irrigation actuators, but only sensors like rain or leaf moisture sensors) in order to provide accurate data on the monitored green areas, thus enhancing the accuracy of the decisions made on the irrigation.

## 4. Campus Radio Channel Analysis

### 4.1. Radio Analysis Characteristics

This section introduces the radio planning tasks performed prior to the deployment of the smart irrigation system. Since the considered outdoor scenario is especially complex in terms of radio propagation, a precise analysis of radio wave propagation must be performed.

The main factor that makes the scenario so challenging is the underground wireless channel, which is a heterogeneous layer of soil, rocks, water, and organic matter, among others. Moreover, the underground channel can also vary depending on the location and weather conditions. In fact, electromagnetic wave propagation through soil and rock has been previously studied extensively for ground-penetrating radar applications [54,55,56]. However, a comprehensive channel model for the underground environment does not exist yet.

The main problems that impact communications when using electromagnetic waves in an underground environment are the following: extreme path loss, reflection/refraction, multi-path fading, reduced propagation velocity, and noise. A detailed description on the impact of these phenomena in the underground communication can be found in [57].

It must be noted that electromagnetic wave propagation depends on the dielectric constant of medium materials, which, in a smart irrigation scenario, is a heterogeneous mixture of air, water, and soil. If the porosity of the soil particles is high, propagation will be better due to the higher amount of air present in the medium. On the contrary, if the presence of water in the soil is high, electromagnetic propagation in the medium will be worse. Thus, particles of sand, clay, and silt of the soil impact communications performance. Moreover, node burial depth also impacts electromagnetic propagation [58,59].

Due to the previously mentioned issues, electromagnetic wave propagation in near-ground and underground environments is challenging. To tackle them, this work makes use of an in-house deterministic three-dimensional ray launching (3D-RL) simulation tool together with an intensive measurement campaign for characterizing the wireless channel within the scenario under analysis.

The used algorithm is based on Geometrical Optics (GO) and the Uniform Theory of Diffraction (UTD), making use of the 3D environment to efficiently launch rays from the transmitter source. Thus, the algorithm launches hundreds of rays from the transmitter source with a predetermined angular separation. Such rays are traced in the scenario according to the presence and distribution of obstacles. Along their trajectory, when they encounter an obstacle, electromagnetic phenomena like reflection, refraction, and diffraction are considered. The 3D-RL principles of operation and the different considered phenomena are represented schematically in Figure 8 for a transmitter located underground.

The used RL technique is also referred in the literature as brute-force Ray Tracing (RT), ‘shooting and bouncing rays’ or beam-launching method [60,61,62]. The technique assumes a space discretization in the simulated scenario, which limits field prediction accuracy and depends on the scenario dimensions. The optimal space discretization for different scenarios has been obtained by experimental measurements in [63]. A detailed description of the used 3D-RL algorithm can be found in [64]. Its validation has been performed for different applications such as vehicular communications [65], smart cities [66], or in homogeneous vegetation environments [67]. It is worth noting that, although the employed 3D-RL algorithm has inherent constraints which lead to estimation errors, mainly the limitation to create the objects present within the scenario by tetrahedrons, and the non-application of the scattering effect due to rough surfaces, the obtained simulation results are very accurate, and, therefore, the approximation valid.

### 4.2. 3D-RL Scenario

Before deploying IoT nodes on a real scenario, it is worth realizing that a smart irrigation system may be impacted by certain limitations. First, the position of the nodes on the ground depends on the access to water supply and on the surface to irrigate. Moreover, the deployment of the LoRaWAN gateway can be optimized based on radio planning analysis. To contemplate such factors, the scenario was modeled in 3D with the 3D-RL tool as it can be observed in Figure 9 (this scenario is actually part of a larger scenario created and used previously in [4]).

To obtain accurate RF power distribution estimations, the simulation scenario was created as detailed as possible, thus including the most relevant elements that may influence electromagnetic propagation (e.g., buildings, roads, trees, and vehicles) and their specific material properties (permittivity and conductivity, as indicated in Table 4). Then, the input parameters of the 3D-RL tool were carefully selected, including the operation frequency, the transmitter, and receiver radiation patterns, the number of permitted reflections and the ray angular/spatial resolution. In addition, in order to obtain accurate simulation results in a limited amount of time, it was necessary to set correctly the number of reflections and the angular/spatial resolution parameters through a proper analysis [63]. The resulting parameters for the scenario under analysis are indicated in Table 5.

### 4.3. Simulation Results

The simulated nodes and gateway locations were virtually placed on the ground on the positions depicted in Figure 10, following the deployment structure previously depicted in Figure 1. In Figure 10, the red circles represent irrigation node locations, while purple circles represent gateway locations. Eight different irrigation node potential locations were analyzed and simulated when exchanging data with a gateway that could be positioned into three different locations. Two of the gateway locations were outdoors (locations 9 and 10, which have a height of 1.8 m), while the other was indoors (location 11, at a height of 5.5 m). The outdoor locations were chosen by considering aspects such as ease of installation (e.g., access to electrical power) and the existence of line of sight with respect to most of the deployed IoT nodes and to the surrounding buildings. The gateway indoor location was selected to be inside one specific building due to its challenging outside enclosure (such a building can be observed on the picture on the left of Figure 2), which is covered by an external metal mesh that acts as a sort of Faraday cage, thus blocking electromagnetic waves partially.

Since the 3D-RL tool provides the RF power distribution for the whole volume of the scenario, it is possible to optimize node and gateway deployment locations. Considering that the developed smart irrigation nodes provide a sensitivity of –148 dBm (both for 433 MHz and 868 MHz), deterministic estimations of the received RF power level at each gateway position can be obtained. Thus, the validity of the proposed locations can be inferred.

Before showing the obtained simulation results, it is important to mention that the created scenario was previously validated by means of empirical results obtained during a measurement campaign [4]. The obtained results showed a very good agreement between measurements and 3D-RL estimations, obtaining a mean error of 0.53 dB and a standard deviation of 3.39 dB.

The results obtained for the 868 MHz and 433 MHz transceivers are presented in Figure 11 and Figure 12, respectively. Specifically, such figures show bi-dimensional planes of the RF power distribution estimations for all the eight smart irrigation nodes at the ground level. As expected, 433 MHz propagation is better in the same conditions. This is due to the higher transmitted power and to its lower radio propagation losses in comparison to the higher 868 MHz frequency.

When the gateway is located at position 9, irrigation nodes 2 and 8 obtain their worst results due to the shadowing created by the building. Even in such a situation, due to the low sensitivity values of both 433 and 868 MHz devices, the received power level is higher than the sensitivity threshold (−148 dBm). This fact was corroborated during the empirical measurement campaign (described later in Section 5).

In the case of placing the gateway into position 10, the worst results are clearly obtained by irrigation node 2, again due to the shadowing created by the building. Like in the previous case, the low sensitivity of LoRaWAN devices facilitates wireless communications also in cases when the shadowing effect due to buildings absorbs a great amount of energy of the propagated wave.

The third analyzed gateway location corresponds to position 11, which is inside the building, at a height of 5.5 m from the street ground level. Figure 13 and Figure 14 show estimations of the RF power level distribution at a height of 5.5 m for 868 MHz (Figure 13) and 433 MHz (Figure 14) transceivers (note that, for this simulation, the irrigation nodes remained on the ground). Like in the previous simulations, higher RF power levels are observed for 433 MHz due to the higher transmitted power level and the lower radio propagation losses. However, in comparison to the results obtained for gateway positions 9 and 10, the results obtained in position 11 are worse. As it can be observed, position 11 is inside blue areas or next to them. This means that the received RF power is significantly lower in comparison to the other two gateway positions, which, in some cases, can lead to packet losses and even to a wireless communication outage. This behavior is confirmed by the empirical measurements presented in the next section.

## 5. Experiments

An empirical measurement campaign was carried out to validate the simulation results detailed in the previous section and to determine node performance on the selected scenario. Thus, signal strength measurements from LoRa and LoRaWAN IoT nodes were collected by gateways when exchanging data from different potential locations. The next subsections detail the characteristics of the used hardware, describe the performed tests, and analyze the obtained results comparing them with the ones obtained by the 3D-RL tool.

### 5.1. LoRa/LoRaWAN Smart Irrigation Testbeds

In order to determine the performance of LoRa and LoRaWAN for the deployment of smart irrigation IoT nodes, two different testbeds were created: one based on LoRaWAN that works at 868 MHz and another one that uses LoRa at 433 MHz.

Figure 15 shows an 868 MHz LoRaWAN-based IoT node and its components. Specifically, the node is based on a WisTrio RAK5205 LPWAN tracker, which makes use of a Semtech SX1276 LoRa transceiver and an STM32L1 microcontroller. The main specifications of the node are indicated in Table 6. Regarding the 433 MHz node, it is based on a Heltec LoRa 32 v1 board (as it can be observed in Figure 16) and its main specifications are detailed in Table 7.

In spite of having different microcontrollers (which essentially affects power consumption), both IoT nodes make use of the same radio module at different frequency bands. Moreover, it is worth mentioning that the RAK5205 board comes with a built-in GPS and different environmental sensors, while the used 433 MHz LoRa board uses an ESP32 as a main controller, which also embeds WiFi and Bluetooth transceivers.

Finally, with respect to the gateways of the testbeds, whose main characteristics are summarized in Table 8, they make use of the following hardware:868 MHz testbed gateway: a RAK7258 LoRaWAN gateway was used. Such a gateway embeds a Semtech SX1301 LoRaWAN transceiver able to provide full 8-channel communications. The gateway also embeds a Mediatek MT7628 System on Chip (SoC), 128 MB of RAM and WiFi and Ethernet transceivers.433 MHz testbed gateway: a point-to-point connection with two 433 MHz LoRa boards was created. Thus, both 433 MHz nodes made use of the same essential hardware (a Heltec LoRa 32 v1 board), but one acted as an IoT node and the other one as a gateway. As it can be observed in Table 8, the main difference between both nodes is that the gateway made use of a 5 dBi omnidirectional antenna, while the IoT node used a 1 dBi coil antenna.

### 5.2. Performed Tests

The tests consisted of transmitting packets from every IoT node outdoor location to the respective gateway (depending on the link quality, between none and roughly 60 packets were received). It is important to note that, since different protocols were used for each testbed, the considerations for measuring their performance and signal strength differ slightly:LoRaWAN is more complex than LoRa. By default, most LoRaWAN transceivers make use of an adaptive algorithm (Adaptative Data Rate, ADR) when transmitting, which implies that their spreading factor and transmission power are adjusted dynamically. In the performed tests, it was observed that, after the transmission of the first packet, the spreading factor was adjusted to a value that remained static for the rest of the transmissions. Likewise, there were no variations in the power level, which was always set in level 1, which implies a transmission power of 14 dBm.In the case of LoRa, the ADR algorithm is not executed, so spreading factor and transmit power values remain static and thus independent from the communications conditions. During the tests, the maximum allowed transmission power for the EU433 band was used, which corresponds to a spreading factor of 12 and a power transmission of 20 dBm.

Regarding the measurement locations, they were the ones analyzed with the 3D-RL simulator and depicted in Figure 1. For every IoT node position, the node was placed on the ground or underground (at a burial depth of roughly 10 cm; one of the moments of the burying process is illustrated in Figure 17) depending on the experiment. Regarding the gateway, during the measurements in position 11, the gateway was located indoors, next to a window at a 5.5-meter height above the ground level. In the case of positions 9 and 10, the gateway was outdoors and its antenna was placed in a tripod, as it is shown in Figure 18.

At every measurement location, the Received Signal Strength Indicator (RSSI) values were collected for the node-to-gateway communications. It is important to note that RSSI depends on diverse factors [69], so the results and conclusions presented in the next section cannot be extrapolated directly to other scenarios and, therefore, they should be considered as an example of the validation of the appropriateness of the proposed smart irrigation system design approach and on the usefulness of performing simulations with the used 3D-RL tool prior to a real hardware deployment.

### 5.3. Results

The measurement campaign provided a lot of information regarding the performance on the wireless link for all the eight irrigation nodes and the three gateway potential positions. Figure 19 shows a summary of the results obtained with an operating frequency of 868 MHz, for both the near-ground (i.e., on the ground) and underground locations of the irrigation nodes. The same kind of measurements but for the 433 MHz nodes are shown in Figure 20. As it can be observed, when an irrigation node is underground, the RSSI measured by the gateways is generally lower, which is expected due to the losses of the ground. Nonetheless, in most cases, the difference between near-ground and underground transmissions is not critical, due to the burial depth of the underground nodes (10 cm). It is worth noting that, when the gateway was located at position 11 (i.e., inside the building), it collected the lowest RSSI levels from all the irrigation nodes. Moreover, it is the only gateway position where some packets were lost (in fact, no packets were received from irrigation nodes 1, 2, and 3), showing the worst performance of all gateway positions. Thus, the obtained results agree with the estimations obtained by the 3D-RL algorithm and previously described in Section 4.3.

In order to gain insight into the obtained measurements, Figure 21 presents, as an example, the RSSI received at the three potential gateway locations for each 868 MHz irrigation node position for the near-ground case. Again, it can be observed how the signal sent by irrigation nodes 1, 2, and 3 does not reach successfully the gateway when it is located at position 11. Moreover, the RSSI received by the gateway at position 11 is the lowest of all the cases.

Regarding positions 9 and 10, the RSSI collected by the gateways depends on the distance between them and the specific transmitting irrigation node, as well as on the presence of obstacles like buildings or cars (as previously depicted and modeled in Figure 10). For example, for irrigation node 8, the gateway in location 10 received much better quality signal than the one in location 9 due to the shorter distance and also the shadowing effect caused by the near-by building. On the contrary, in the case of irrigation node 6, the gateway at location 9 received better signal levels than the one at 10, even despite location 9 being further from the irrigation node. This effect was due to the fact that several vehicles (which are mainly metallic) were parked between node 6 and location 10, thus blocking partially radio waves and leading to higher RF losses.

All the measurements agree with the simulation results presented in Section 4.3, both at 433 MHz and 868 MHz, and lead to the same conclusions: position 11 is the worst position for placing the communications gateway. On the other hand, locations 9 and 10 present different performance in terms of RSSI depending on the irrigation node positions, but in both cases the performance in terms of lost packets is the same (zero packets were lost for near-ground and underground nodes, both for 433 MHz and 868 MHz), so such positions are valid for the proposed application. In any case, if larger areas were monitored and controlled by irrigation nodes, the better radio propagation achieved at 433 MHz could lead to the use of such nodes instead of the ones that operate at 868 MHz. Furthermore, once the 3D-RL simulator has been validated, new RF power distribution estimations could be obtained in order to assess the impact of transmitting with lower power levels (for energy saving), which will impact the RSSI and, therefore, the possibility of losing packets due to Signal-to-Noise Ratio (SNR) and to the sensitivity threshold of the devices. In the same way, new gateway and irrigation node positions could be easily analyzed without the necessity of new extensive measurement campaigns.

## 6. Conclusions

This article proposed a smart irrigation system based on LoRa/LoRaWAN IoT nodes and gateways with a novel communications architecture able to exchange data with local fog computing nodes and with a remote cloud in a campus scenario. The prototype of the proposed IoT system was developed and described in detail so as to allow future developers to replicate it easily. In order to validate the proposed system and assess the potential locations for the nodes to be deployed, an in-house 3D-ray launching radio-planning tool was used in conjunction with an accurate 3D model of the simulated scenario (a university campus). In such a scenario, simulations and empirical measurements were carried out for 433 MHz LoRa-based and 868 MHz LoRaWAN-based testbeds. The obtained results validate the accuracy of the used radio planning tool (the results obtained in this scenario showed a very good agreement between measurements and 3D-RL estimations, obtaining a mean error of 0.53 dB and a standard deviation of 3.39 dB), which allows for estimating the RF power level distribution of the involved nodes and gateways for the whole volume of the scenario. Thus, the potential coverage of the different areas can be identified prior their deployment. The obtained radio propagation results also remark, on one hand, the negative effect that some architectural elements of a building (such as the metallic facade in this case) may have in terms of radio propagation, and, on the other hand, that the impact of underground communications in signal attenuation may not be critical if the burial depth is shallow (10 cm in this case). The conclusions obtained show that the proposed tool provides useful guidelines for future developers on the design of smart irrigation systems in similar large scenarios. Further work will add additional intelligence (e.g., predictive models with AI to fine-grained the amount of water) to the IoT nodes layer and will study in depth the effect of burial depth in LoRaWAN underground wireless channels.

## Figures and Tables

**Figure 1 sensors-20-06865-f001:**
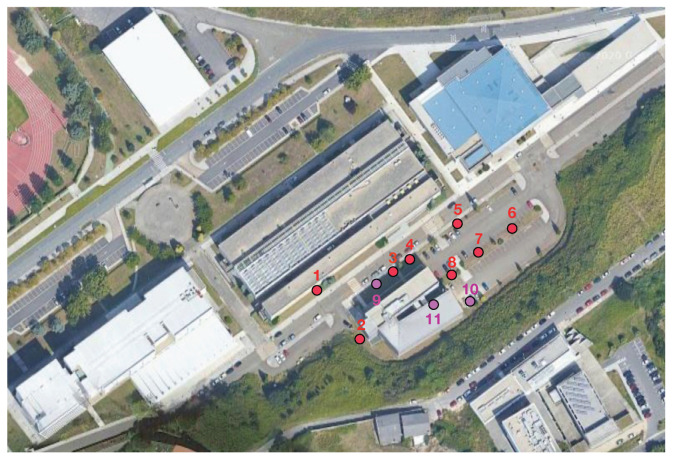
Analyzed scenario with deployed nodes (red circles) and gateway locations (purple circles) (Map source: ©2020 Google).

**Figure 2 sensors-20-06865-f002:**
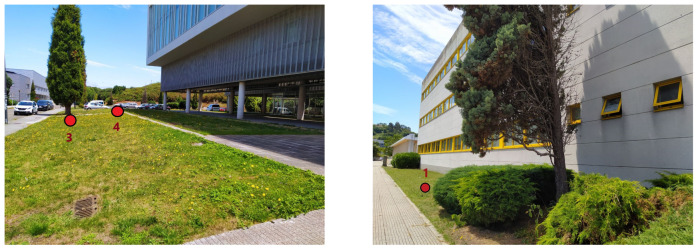
Areas where smart irrigation nodes 3 and 4 (**left**) and 1 (**right**) were located.

**Figure 3 sensors-20-06865-f003:**
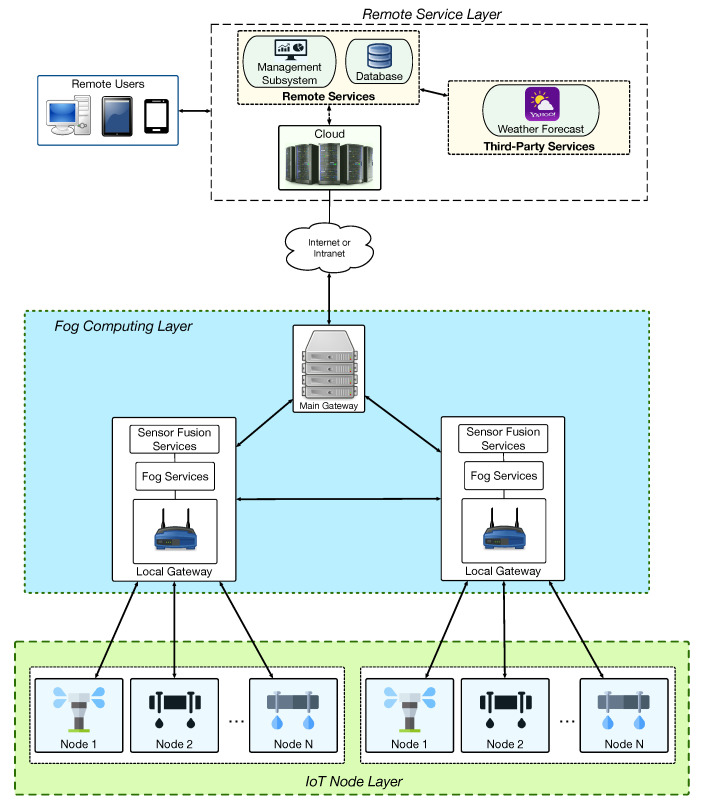
Designed communications architecture of the proposed system.

**Figure 4 sensors-20-06865-f004:**
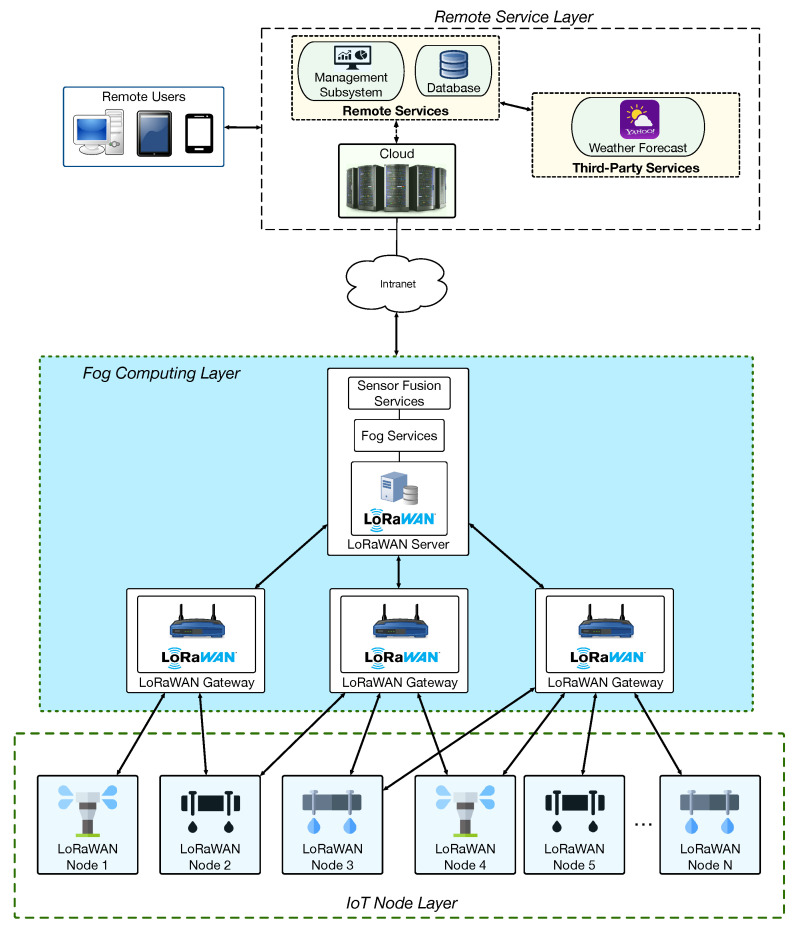
Implemented communications architecture.

**Figure 5 sensors-20-06865-f005:**
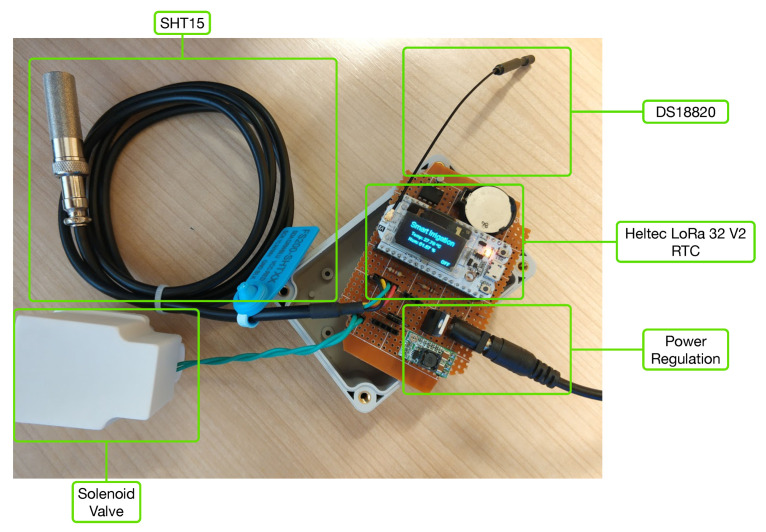
Disassembled smart irrigation node and its internal components.

**Figure 6 sensors-20-06865-f006:**
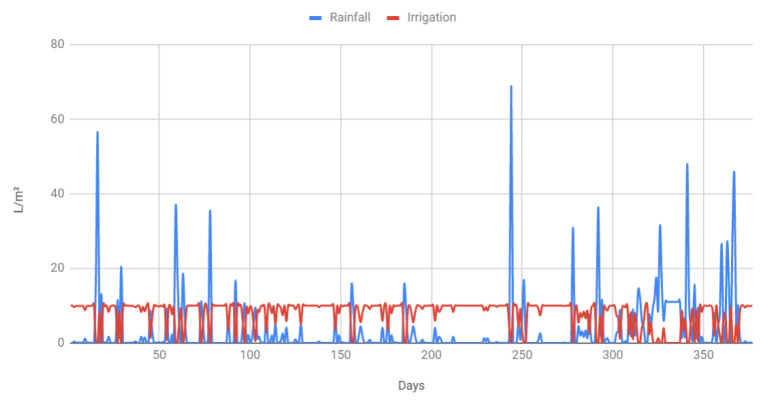
Estimated irrigation per day during a year when depending on rain forecasts.

**Figure 7 sensors-20-06865-f007:**
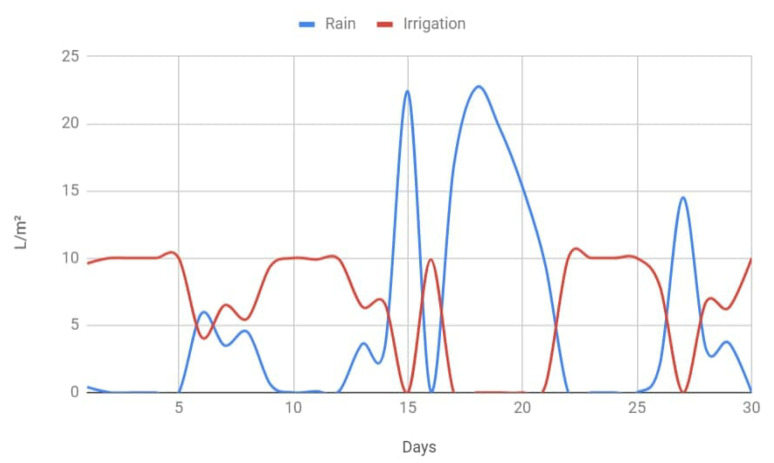
Estimated irrigation per day during the month of September 2020 when depending on rain forecasts.

**Figure 8 sensors-20-06865-f008:**
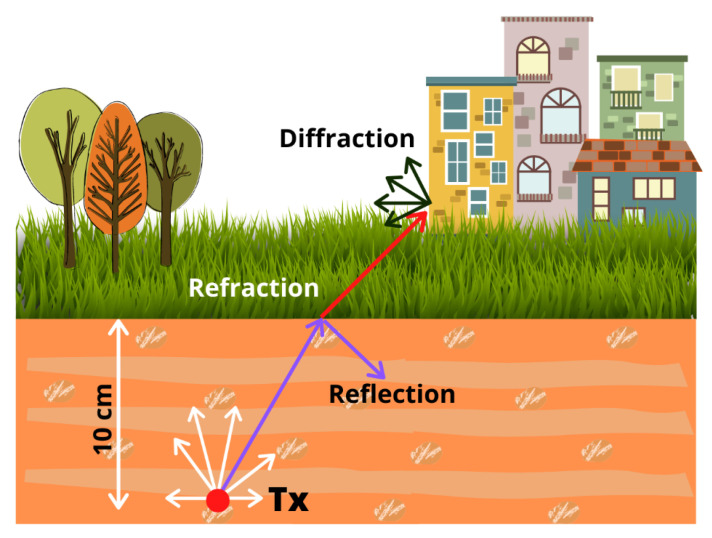
Operation principles and propagation phenomena of the used 3D-RL algorithm.

**Figure 9 sensors-20-06865-f009:**
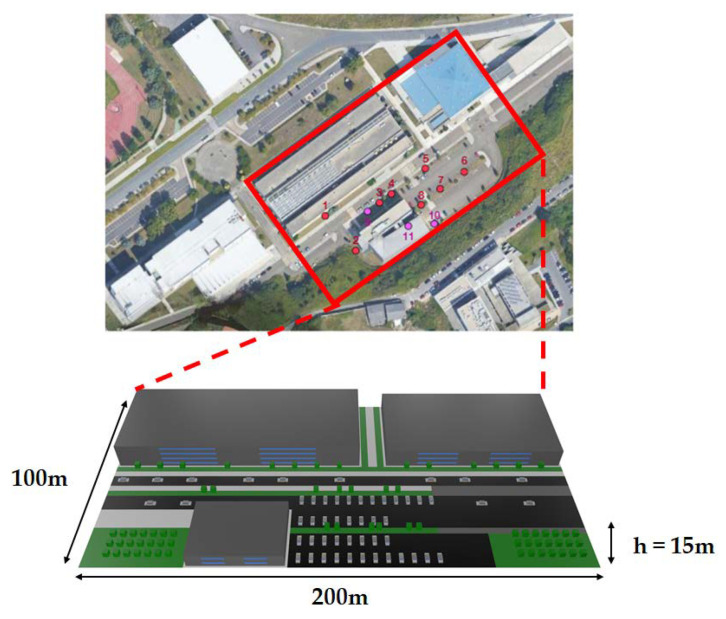
Scenario created for 3D-RL simulations (Map source: ©2020 Google).

**Figure 10 sensors-20-06865-f010:**
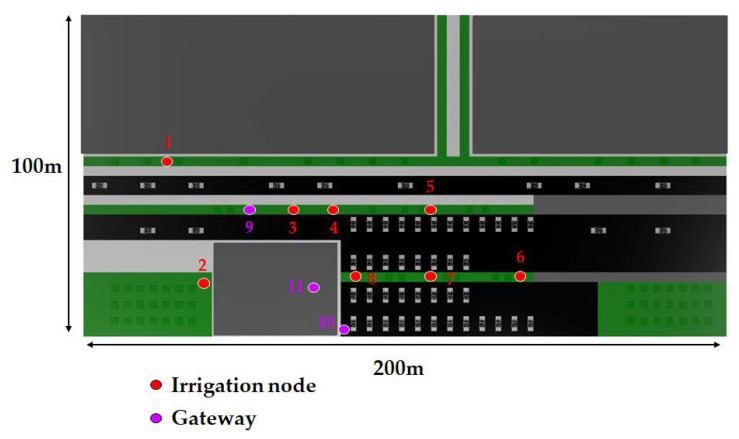
Schematic upper view of the deployment of the irrigation nodes and the gateways.

**Figure 11 sensors-20-06865-f011:**
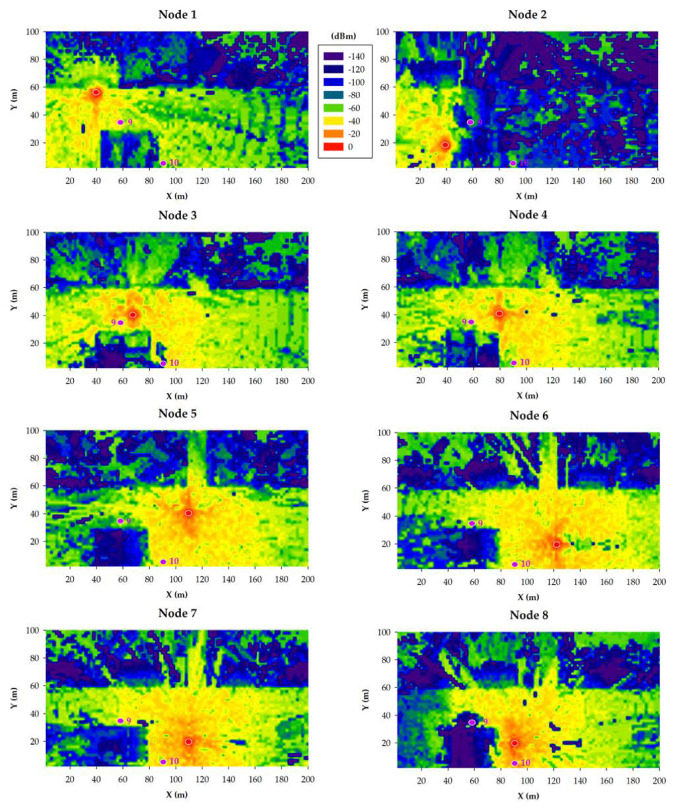
868 MHz simulation results: RF power level distribution in a bi-dimensional plane on the ground level.

**Figure 12 sensors-20-06865-f012:**
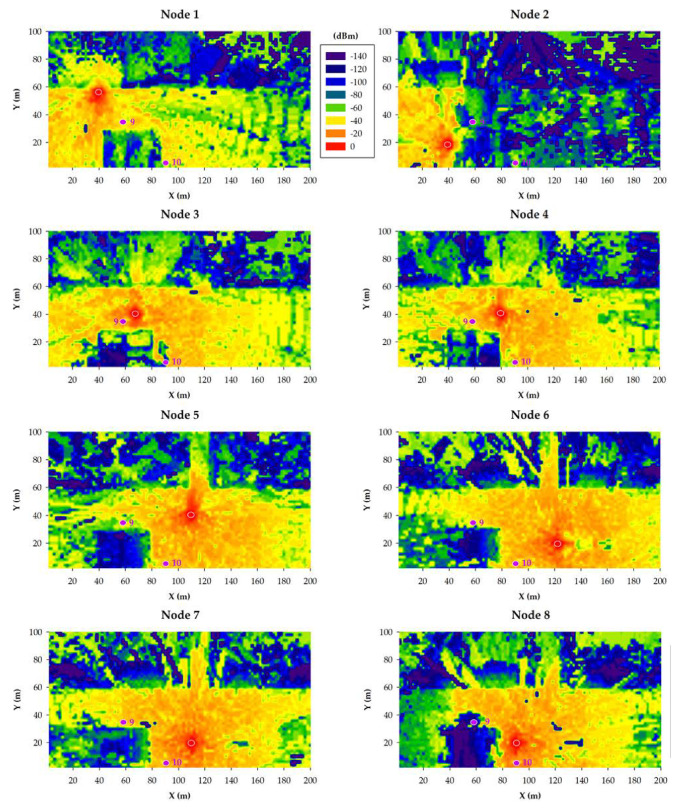
433 MHz simulation results: RF power level distribution in a bi-dimensional plane on the ground level.

**Figure 13 sensors-20-06865-f013:**
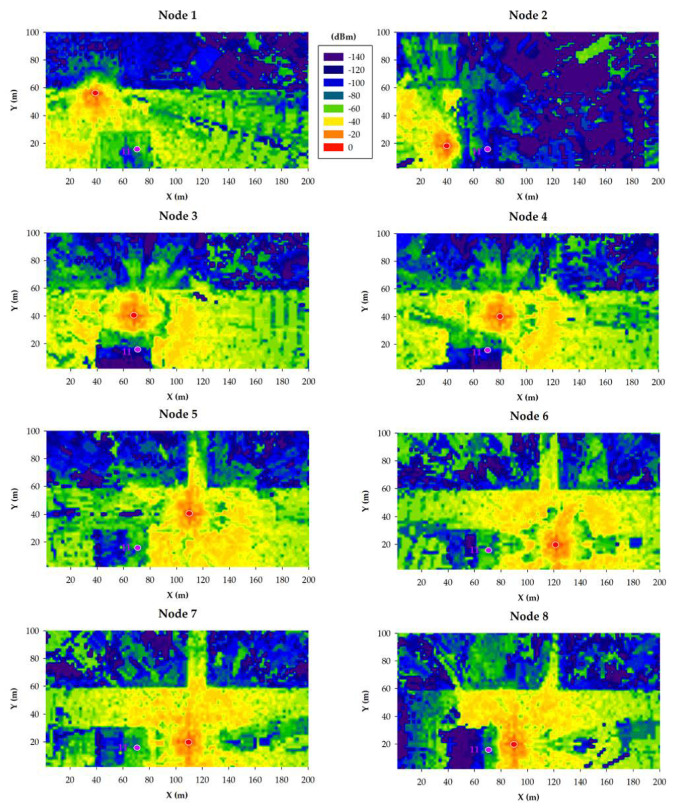
3D-RL simulation results at 868 MHz: RF power level distribution in a bi-dimensional plane at the height of 5.5 m.

**Figure 14 sensors-20-06865-f014:**
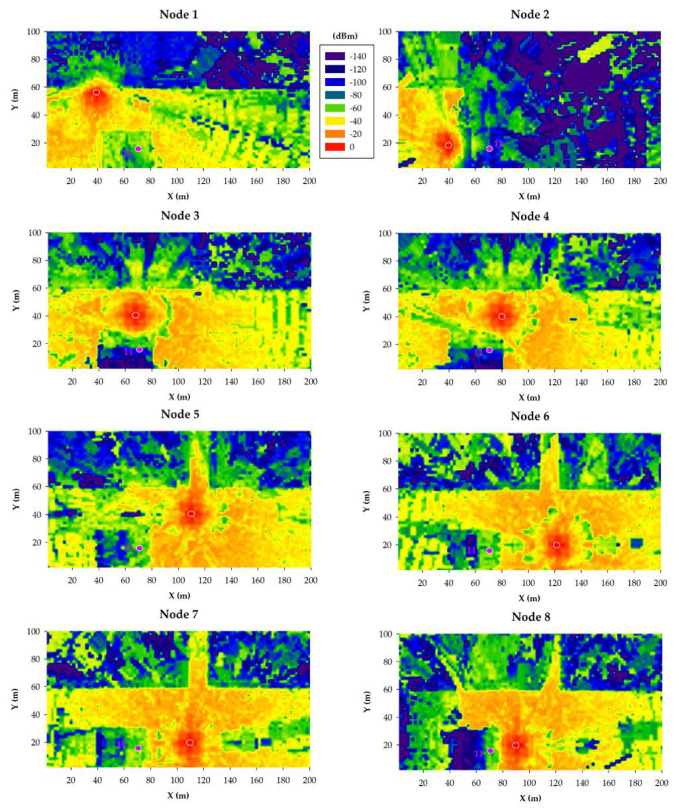
3D-RL simulation results at 433 MHz: RF power level distribution in a bi-dimensional plane at the height of 5.5 m.

**Figure 15 sensors-20-06865-f015:**
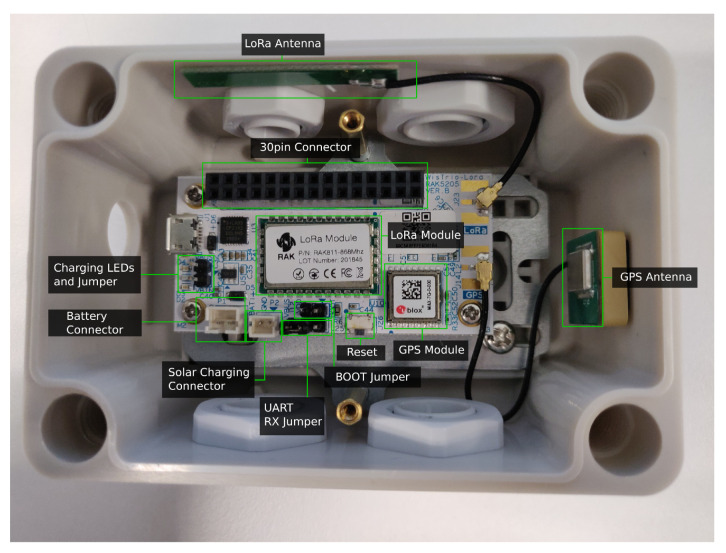
Components of the 868 MHz LoRaWAN node.

**Figure 16 sensors-20-06865-f016:**
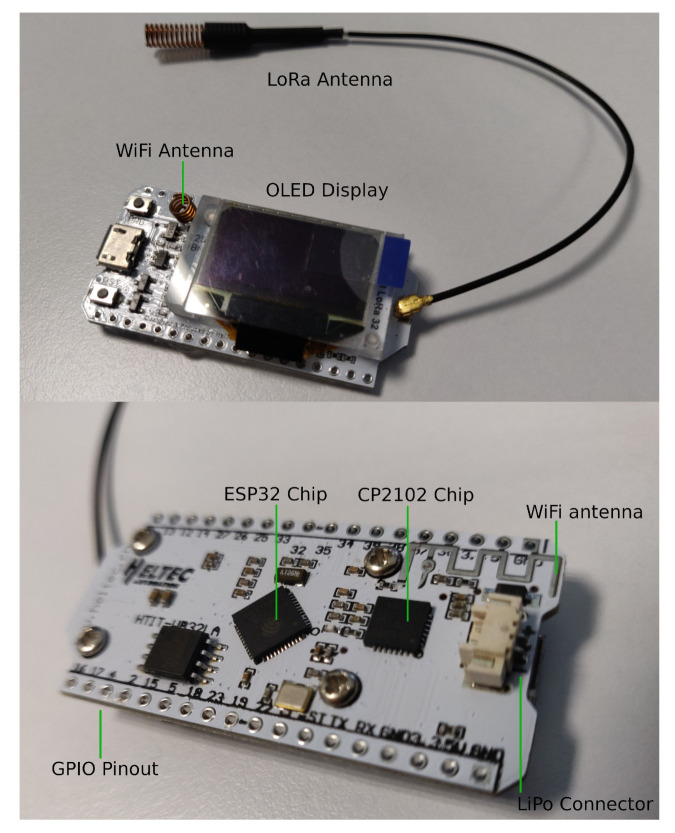
Components of the 433 MHz LoRa node.

**Figure 17 sensors-20-06865-f017:**
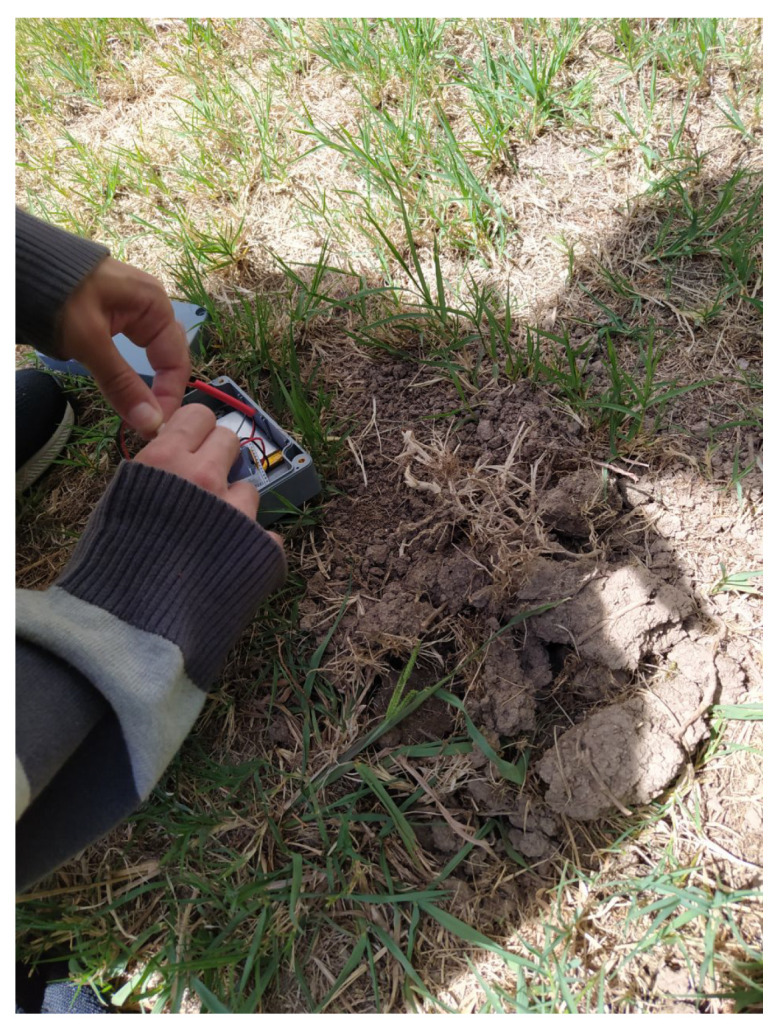
Deployment of one of the smart irrigation nodes.

**Figure 18 sensors-20-06865-f018:**
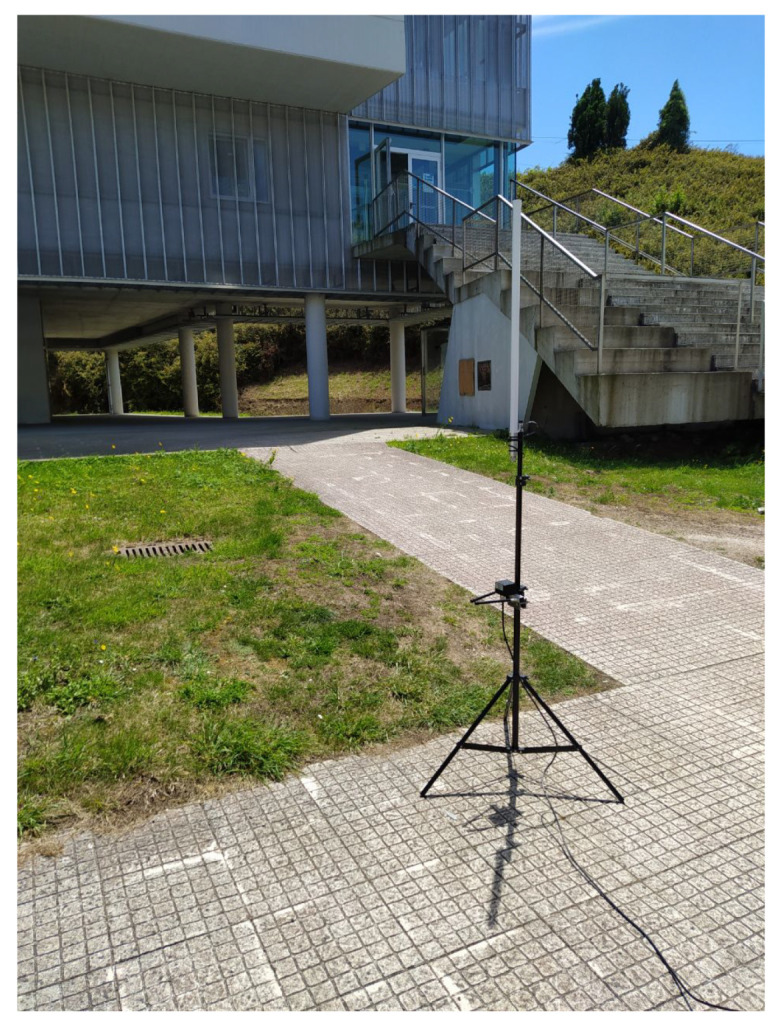
Location of the gateway antenna during the experiments at position 9.

**Figure 19 sensors-20-06865-f019:**
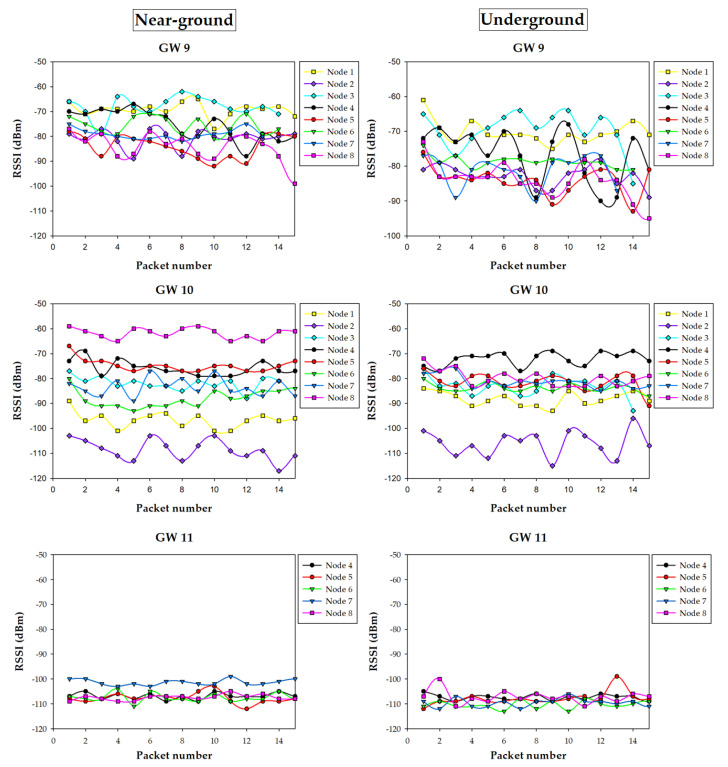
Summary of the measurements at 868 MHz.

**Figure 20 sensors-20-06865-f020:**
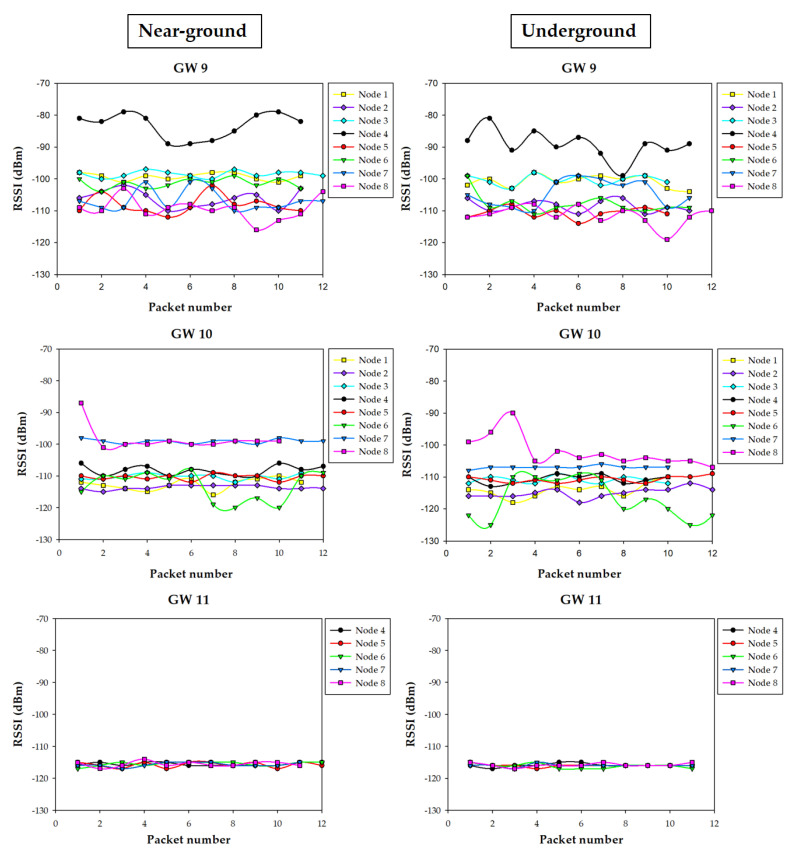
Summary of the measurements at 433 MHz.

**Figure 21 sensors-20-06865-f021:**
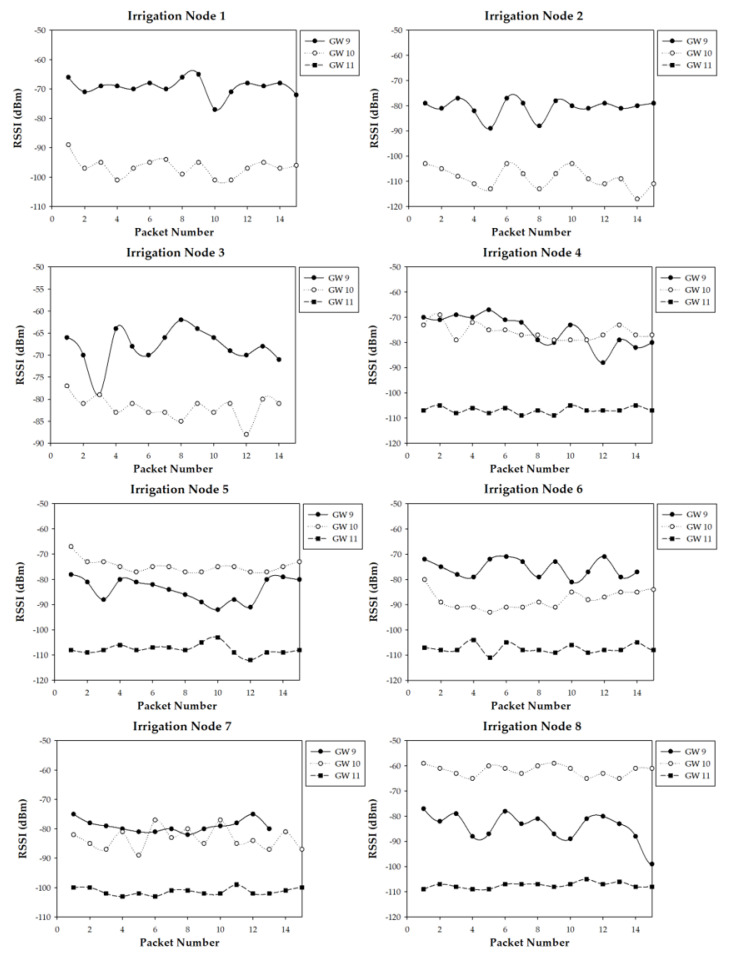
RSSI received by the gateways from the eight irrigation nodes (near-ground) that operate at 868 MHz.

**Table 1 sensors-20-06865-t001:** Overview of the main aspects of smart irrigation systems.

Objective	Water Distribution	Scheduled Irrigation	Actuators	Monitoring Parameters	IoT Nodes (Microcontrollers, SBCs)	Communication Technologies	Cloud Platforms	Performance Indicators	Advanced Features
Wireless Sensor Network	Flood irrigation	Estimated needs	Motor/Pumps	Air temperature	Arduino	Cellular 4G/5G	FIWARE	Expenditure in irrigation (€/m^3^ by year)	Machine learning (AI)
Control System	Spray irrigation	Ad hoc	Valves	Water level	Node MCU	Bluetooth, BLE	Thingspeak	Irrigation (m^3^/year)	Thermal imaging
Decision Support System (DSS)	Drip irrigation		Sprinkler	Water conductivity	Arduino Mega	RFID		Energy consumption	Energy harvesting
Testbed	Nebulizer irrigation			Water temperature	Raspberry Pi	ZigBee			Remote sensing
	Others (e.g., robot)			Rain	Intel Galileo Gen-2	Z-Wave			Fuzzy logic
				PH (soil or water)	ATmega series	Thread		
				Humidity	MSP series	WiFi 802.15b/g/n/ac/ah		
				Soil moisture	STM series	LoRaWAN		
				Plants heigh		SigFox		
				Leaf wetness		NB-IoT		
				Weather forecast		LTE-M		
				Wind		MIOTY		
						RPMA		

**Table 2 sensors-20-06865-t002:** Comparison of some of the features of the most relevant smart irrigation systems and the proposed solution.

Reference	System Type	Location	Covered Area (km^2^)	Communication Technologies	Sensors and Actuators	Fog/Edge Computing Support
Khan et al. [12]	DSS	Orange orchard	Small area	Xbee 802.15.4 module	Soil moisture, temperature, air humidity, and leaf wetness	No
Togneri et al. [13]	IoT ML-based framework	Spain and Brazil (different needs)	-	LoRaWAN/4G	Moisture sensor probes	Fog support (no implementation)
Gloria et al. [18]	WSN for water saving	Small garden, Instituto Universitario de Lisboa	-	LoRa, WiFi, BLE	Temperature, humidity and soil moisture	No
Usmonov et al. [19]	Drip irrigation testbed	No practical deployment	-	LoRaWAN, WiFi	No sensors, but supports up to four actuators per node)	No
Zhao et al. [20]	Testbed, Proof-of-Concept	Urban environment	Up to 8 km (covering an area of up to 2 km^2^	LoRaWAN	Actuators only (water pump, mist sprayer)	No
Citoni et al. [21]	Review state-of-the-art	-	Large-scale deployments	LoRaWAN	-	-
Proposed Solution	IoT smart irrigation system simulation and empirical validation	University campus	7500 m^2^	LoRa, LoRaWAN	Each node has soil moisture/temperature and air temperature sensors, and a solenoid valve	Yes

**Table 3 sensors-20-06865-t003:** LoRa/LoRaWAN main specifications (Europe).

Parameter	Value
Frequency band	EU433 (433.05–434.79 MHz), EU864-870 (863–870 MHz)
Channels	10
Channel bandwidth	125 KHz or 250 KHz
Transmission power	14 dBm
Max output power	20 dBm
Spreading factor	7–12
Data rate	250 bps–5.5 kbps
Link budget	155 dB
Range	5 km (urban), 15 km (suburban), 45 km (rural)
Topology	star
Battery lifetime	years
Power efficiency	very high
Interference immunity	very high
Scalability	yes

**Table 4 sensors-20-06865-t004:** Material properties for the 3D-RL simulations.

Parameters	Permittivity (*ε_r_*)	Conductivity [S/m]
Air	1	0
Glass	6.06	10^−12^
Concrete	5.66	0.142
Metal	4.5	4 × 10^7^
Rubber	2.61	0
Tree foliage	[68]	[68]
Tree trunk	1.4	0.021
Grass	30	0.01

**Table 5 sensors-20-06865-t005:** 3D-RL simulation parameters.

Parameter	EU868 Band	EU433 Band
Frequency	868.3 MHz	433.5 MHz
Power	16 dBm	20 dBm
Reflections	6	6
Cuboids	2 m × 2 m × 1 m	2 m × 2 m × 1 m
Rays resolution	1°	1°
Antenna	Monopole, 5.8 dBi	Monopole, 5 dBi

**Table 6 sensors-20-06865-t006:** Main specifications of the 868 MHz LoRaWAN node.

Characteristic	Value
Protocol	LoRaWAN 1.0.2
RF Module	SX1278
RF Sensitivity	−148 dBm
Maximum Tx Power	20 dBm–100 mW
Antenna	PCB dipole 1 dBi gain
Maximum Link Budget	168 dBi
CPU	ARM Cortex-M3 (32 bits)
Clock Speed	32 MHz
RAM	32 KB
ROM	128 KB

**Table 7 sensors-20-06865-t007:** Specifications of the 433 MHz IoT node.

Characteristic	Value
Protocol	LoRa
RF Module	SX1278
RF Sensitivity	−148 dBm
Maximum TX Power	20 dBm–100 mW
Antenna	Coil Antenna (dipole) aprox. 1 dBi gain
Maximum Link Budget	168 dBi
CPU	ESP32 - Tensilica LX6 dual core
Clock speed	240 MHz (maximum)
RAM	520 KB
ROM (Flash)	8 MB

**Table 8 sensors-20-06865-t008:** Main characteristics of the LoRa/LoRaWAN gateways of the testbeds.

Band	868 MHz - EU868	433 MHz - EU433
Protocol	LoRaWAN	LoRa
RF Module	SX1301	SX1278
RF Sensitivity	−142 dBm	−148 dBm
Transmission TX Power	14 dBm	20 dBm
Receiver Antenna	Omnidirectional 5.8 dBi gain	Horizontal-Polarization Omnidirectional 5 dBi gain
Sender Antenna	1 dBi PCB Antenna	1 dBi Coil Antenna
Spread Factor	Adaptative (from 9 to 12)	12
Channel Bandwidth	125 KHz	125 KHz

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
