# Peer review of "Design, Implementation, and Empirical Validation of an IoT Smart Irrigation System for Fog Computing Applications Based on LoRa and LoRaWAN Sensor Nodes†"

_sensors, 2020, doi:10.3390/s20236865_

Round 1

Reviewer 1 Report

It could be better of the authors complete the manuscript with some constraints by applying this method.

The test case or case study covers a small area, only 2 km2. Any comments or suggestions in case you have a much larger area to be applied?

How about the possible effect of the digital waves in case near to the area, other irrigation or drainage systems apply also any digital or smart system? May be good if possible to discuss a bit the possible impacts of other smart systems in the surrounding area.

Author Response

Dear Sir/Madam,

The authors would like to thank the reviewer for his/her valuable comments, which have certainly helped us to improve the manuscript. Please find attached our detailed responses to the comments. In order to ease the labor of the reviewers we have colored in red the differences with the previous version of the article.

Best regards.
The authors.

Reviewer 2 Report

This paper develops a smart irrigation system which is based on the LoRA/LoRaWAN architecture. It builds a three layer system consisting of three layers, the IoT node Layer, the fog computing layer and Remote Service Layer. Details are also provided regarding the implementation of all three layer. The paper also describes the heterogeneous radio channel irrigation environment concentrating on some various attempts made so far. They also used a 3D-RL tool to study the performance of various factors, such as RF power distribution estimations,  dominating the simulated irrigation/environmental scenario. All the aspects are described quite well (similar papers have been also been published), and they are offering some basis for a good review, always necessary before the implementation phase. However they add nothing new regarding the communication or operation of such an implemented system. Perhaps the experiments presented, both at 433MHz and 866 MHz show some not surprising results, but they need to be validated before use in a real wide-area environment, and micro-climatic conditions. Note that the smart irrigation scenario is implemented  in a very small area of 0,75 ha, with very limited number of nodes. Thus, in real terms the conclusions are limited.

PS: Figs 9-12 could be significantly reduced   

Author Response

(The authors gave the same response as above.)

Reviewer 3 Report

In this article, the author designed an IoT smart irrigation system for fog computing applications based on LoRa and LoRaWAN sensor nodes. The idea is very interesting and attractive. It is easy to follow and written well. Of course, I also found several concerns should be handled before publication: Q1. Abstract in this paper should be double-checked and revised. Note Abstract as an independent part of this paper should include the main idea, methods, goals and results. Q2. The use of the two-way arrow in Figure 3 is not clear. For example, weather forecast and terminal service should be one-way data transmission. Q3. The research topic is irrigation system, and the IoT Node Layer should be introduced in detail. Q4. It will be appreciated if the authors can investigate the water saving and economic efficiency of the designed irrigation system. Q5. In the conclusion, you should summarize the research from both theoretical significance and practical significance, and puts forward the shortcoming and future direction.

Author Response

(The authors gave the same response as above.)

Reviewer 4 Report

This article proposed a smart irrigation system based on LoRa/LoRaWAN IoT nodes and gateways to exchange data with local fog computing nodes and with a remote cloud. The work is very novel and interesting. Some minor places:

  1. Figure 3 and 4 are mostly duplicated. It is not necessary to separate them into two figures.
  2. I am particularly interested in the irrigation schedule and the vegetation health. Since it is a "smart irrigation" system, would you please explain further on how smart the irrigation is? Will the irrigation schedule be autonomously changed by the system when the soil, air, precipitation conditions changed?
  3. This system seems requiring a lot of gateways if we want to deploy it in a large cropland. Can you introduce the details about the framework's scalability?

Author Response

(The authors gave the same response as above.)

Reviewer 5 Report

This paper addresses the general problem of drought risk mitigation by designing, implementing and validating a smart irrigation system that can be deployed in urban and suburban scenarios.

More specifically, it focuses on LPWAN solutions such as LoRa and LoRaWAN, fog computing, simulation and empirical RF measurements.

A real scenario based on a university campus is used for on-site experiments, an RF simulator is validated to radio planning and determine optimal location of sensors and gateway. 

The paper is well written, easy to follow and provides a valid contribution.

It is tested and validated in a relatively simple scenario and the results are not astonishing.

It would be great to apply the simulator to a much wider area for optimal allocation of devices under real operating conditions. Afeter the validation, one expect the analysis of a more complex use case.

Is the simulator available as open source?

LoRa and LoraWan are used without a preliminary discussion. I suggest to add a section at the beginning of section 3.3 with introductory material  on such technologies, highlighting the main architecture and clear positioning in the communication stack.

I also suggest to give a look to the Special Issue on Fog, Edge, and Cloud Integration for Smart Environments on ACM TOIT, April 2019. It contains interesting papers that could be referenced.

Figure 14 appears upside down in the file.

Author Response

(The authors gave the same response as above.)

Round 2

Reviewer 2 Report

No comments

Reviewer 3 Report

No more comments

Reviewer 5 Report

The Authors have carefully followed the suggestion of the referee.

The paper has significantly improved, with new parts and details.

One single note: in the short description of LoRaWan clarify the role of  Gateway and Network Server and their main functions. If necessary, add also a figure.

The papaer is now in a good shape to be published.